# A high-throughput integrated biofilm-on-a-chip platform for the investigation of combinatory physicochemical responses to chemical and fluid shear stress

**Ann V. Nguyen**[1⦿], **Arash Yahyazadeh Shourabi**[1⦿], **Mohammad Yaghoobi**[1], **Shiying Zhang**[2], **Kenneth W. Simpson**[2], **Alireza Abbaspourrad**[1]*

1 Department of Food Science, College of Agricultural and Life Sciences, Cornell University, Ithaca, New York, United States of America, 2 Department of Clinical Sciences, College of Veterinary Medicine, Cornell University, Ithaca, New York, United States of America

⦿ These authors contributed equally to this work.

* Alireza@cornell.edu

**Data Availability Statement:** All data files are available at the following DOI: https://doi.org/10.5281/zenodo.6874852.

## Abstract

Physicochemical conditions play a key role in the development of biofilm removal strategies. This study presents an integrated, double-layer, high-throughput microfluidic chip for real-time screening of the combined effect of antibiotic concentration and fluid shear stress (FSS) on biofilms. Biofilms of *Escherichia coli* LF82 and *Pseudomonas aeruginosa* were tested against gentamicin and streptomycin to examine the time dependent effects of concentration and FSS on the integrity of the biofilm. A MatLab image analysis method was developed to measure the bacterial surface coverage and total fluorescent intensity of the biofilms before and after each treatment. The chip consists of two layers. The top layer contains the concentration gradient generator (CGG) capable of diluting the input drug linearly into four concentrations. The bottom layer contains four expanding FSS chambers imposing three different FSSs on cultured biofilms. As a result, 12 combinatorial states of concentration and FSS can be investigated on the biofilm simultaneously. Our proof-of-concept study revealed that the reduction of *E. coli* biofilms was directly dependent upon both antibacterial dose and shear intensity, whereas the *P. aeruginosa* biofilms were not impacted as significantly. This confirmed that the effectiveness of biofilm removal is dependent on bacterial species and the environment. Our experimental system could be used to investigate the physicochemical responses of other biofilms or to assess the effectiveness of biofilm removal methods.

## Introduction

Biofilms are surface-adhered aggregations of bacteria encased in a self-developed extracellular polymeric substance (EPS). EPS functions as a protective shield against environmental stresses, such as temperature variations, fluid shear stress (FSS), changes in ions, pH [1], and it also promotes antibacterial resistance [2]. For this reason, it is difficult to remove these 3D tissue-like

**Funding:** This work was supported in part by the United States Department of Agriculture's National Institute of Food and Agriculture under award number 2019-38420-28975. This work was performed in part at the Cornell NanoScale Facility, a member of the National Nanotechnology Coordinated Infrastructure (NNCI), which is supported by the National Science Foundation Grant NNCI-2025233. The funders had no role in study design, data collection and analysis, decision to publish, or preparation of the manuscript.

**Competing interests:** The authors have declared that no competing interests exist.

structures from their substrates mechanically, and from the chemical point of view, the required dosage of an antibiotic drug to kill bacteria in a biofilm is usually higher than the concentration needed to kill them in the planktonic phase [3, 4]. Biofilms can form and grow on both biotic and abiotic surfaces [5]. They cause infection, contamination, and corrosion [6] in various cases: as dental plaques [7], on medical implants [8], inside lung airways [4], on clinical equipment [9], inside the pipelines of the food, pharmaceutical [10, 11], or water [12] industries. *Pseudomonas aeruginosa* biofilms for instance, are linked to chronic pulmonary infections due to their resistance to both antibiotic treatments and the immune system [13]. As another example, *Escherichia coli* biofilms aggregated on food contact surfaces in industrial sites may contribute to food spoilage and subsequent infections [14].

Physically, fluid shear stress is a key player in biofilm formation and behavior in the real world although it is still poorly understood to date [2, 15]. Not only does the hydrodynamics of the bulk flow affect the biofilm's life-cycle, it can also have an impact on biofilm-chemical interaction [16]. A comprehensive understanding of the combined effect of physicochemical parameters, such as fluid shear stress and antibiotics, on biofilms is a crucial step in the never-ending battle against bacteria [17]. Biofilms are exposed to various magnitudes of FSS in vivo, in piping systems, and also with respect to medical tools [1, 2, 18]. The FSS value may vary from 0–20 dyne/cm$^2$ covering a wide range of biological, biomedical, and industrial applications [4, 12, 19–24]. The vital importance of applying FSS on biofilms shows itself in four types of studies: (1) biophysical studies on biofilms such as the effect of FSS on biofilm morphology [25–27] or the robustness of bacterial quorum sensing [28]; (2) investigating the effect of FSS on biofilm formation and growth [27] like *Staphylococcus aureus* attachment to orthopedic materials in the presence of hydrodynamic flow [21]; (3) mimicking in vivo hydrodynamic conditions to bridge the gap between in vivo and in vitro for biofilm-related drug screening projects [3, 29]; and (4) developing biofilm cleaning/eradication strategies from different biomedical or industrial surfaces [23].

Conventional platforms to study biofilm formation [30], growth [31], antibiotic treatment [32], and flow interaction [33] are either static (microtiter dish biofilm) or dynamic (drip flow reactors) but not high-throughput. Additionally, most of them consume a considerable amount of costly reagents for each test that limits their application in long-run studies [33]. Increasingly, Microfluidics has been used as an emerging tool to overcome these challenges [6, 34]. Beside miniaturizing traditional benchtop assays [35, 36], microfluidics is also a powerful tool being used in cell-based physicochemical studies [37, 38]. Lab-on-a-chip devices for this purpose provide researchers with the online and simultaneous screening of the effect of various fluidic [39–42] and chemical [43] conditions on cells and biofilms. In this regard, integrated microfluidic chips have paved the way for studying the combined effect of hydrodynamic flow and chemical concentration on cells in a high-throughput manner [17, 44]. Specifically, for biofilm-on-a-chip studies, it is of great merit to be able to precisely adjust the fluidic conditions when treating the biofilm with chemical reagents to better simulate real-world conditions on a chip scale [39]. Several physicochemical studies followed the idea of integrating a concentration gradient generator with a perfusion culture chamber to treat the biofilm with various concentrations of chemical reagents at an adjusted FSS value [4, 17, 34, 45]. While their approach is high-throughput from the chemical aspect, from the physical point of view, FSS, these previous studies are limited. Next level designs, therefore, should make biofilm-on-a-chip systems high-throughput from both the chemical and physical aspects.

We have designed and tested a double-layer, integrated microfluidic chip for high-throughput physicochemical studies on biofilms. Our chip contains a two-stage, tree-like CGG and four expanding FSS chambers, and is capable of treating biofilms with four diluted

concentrations of antibiotics while imposing three different FSSs (low, medium, high). Overall, 12 distinct combinations of drug concentrations and FSS magnitudes can be screened simultaneously on our chip. Thus, our chip can potentially address all four categories discussed earlier. We present here the proof-of-concept functionality of our 2-layer physicochemical analysis biofilmchip, 2PAB, by examining the physicochemical effects of two common antibiotics, gentamicin and streptomycin, combined with FSS on separate, established (24 h) biofilms of *Escherichia coli* and *Pseudomonas aeruginosa*, the bacteria most prevalent in biofilm-related infections [46–48].

## Experimental

### Strains and culture conditions

To allow clinical relevance of the findings, we considered bacteria that have a history of developing biofilms and that can cause infections in humans as our model systems. Especially *P. aeruginosa*, which forms antibiotic-resistant biofilms that are difficult to eradicate [48]. RFP-labeled *P. aeruginosa* PA01 and GFP-labeled *E. coli* LF82 from the Simpson lab archive were used in this study. *E. coli* LF82 showed better biofilm forming ability among four strains of Crohn's disease associated bacteria, including invasive and adherent invasive strains (S1 Fig in S1 File). While we did not conduct an in-house study of our pathogens' ability to form biofilms, results previously published have suggested that they are both strong biofilm formers based on their Specific biofilm formation (SBF) index. The SBF index of *E. coli* LF82 was reported to be 1.6 [49] to 2 [50], and of *P. aeruginosa* PA01 was reported to be > 3.5 [51]. Bacteria with SBF >1 are considered to be strong biofilm producers. Bacteria colonies were grown on LB plates with the addition of kanamycin (for GFP- *E. coli*) or ampicillin (for RFP-*P. aeruginosa*) at 37 ˚C. Prior to seeding into the biofilm chip, one colony of each bacteria was inoculated into a fresh LB medium with the respective antibiotics and incubated overnight at 37 ˚C with shaking at 120 rpm.

### Biofilm chip design, fabrication, and characterization

The 2PAB device was fabricated using soft lithography techniques at the Cornell Nanoscience Facility, as described previously [36]. In brief, the device consists of two polydimethylsiloxane (PDMS) layers bonded together; all bacteria and antibiotics will be in contact with the interior PDMS surfaces. The bonded PDMS layers were bonded to a glass plate to provide support to the PDMS layers. The top PDMS layer is patterned with the CGG with depth of 200 μm and the middle PDMS layer is patterned with four biofilm culture chambers with depth of 40 μm. The dimension of the diffusive mixer was 200 μm (width) and the biofilm microchambers had height of 40 μm and different widths (100 μm, high-FSS zone; 400 μm medium-FSS zone; 1000 μm, low-FSS zone) for each FSS zone. The top layer also contains two inlets for culture medium and culture medium containing antibiotics, and an additional port for bacterial seeding and system outlet. The two PDMS layers were fabricated separately and assembled by plasma treatment for 50 s, followed by bonding of the combined PDMS layers to a glass slide. The vertical cylinders were used to help align the two layers. The bonded device was further incubated at 60 ˚C overnight to provide added stability. A Chemyx syringe pump (Stafford, TX) was used to control fluid flow rates in the device. Generation of the desired concentration range in the device was verified by forming a gradient of resazurin at a flow rate of 300 μL/h for 1 hour and quantifying the relative concentration of resazurin in each chamber using fluorescence microscopy and image analysis by ImageJ. Detailed drawings of the chip can be found in Fig 1.

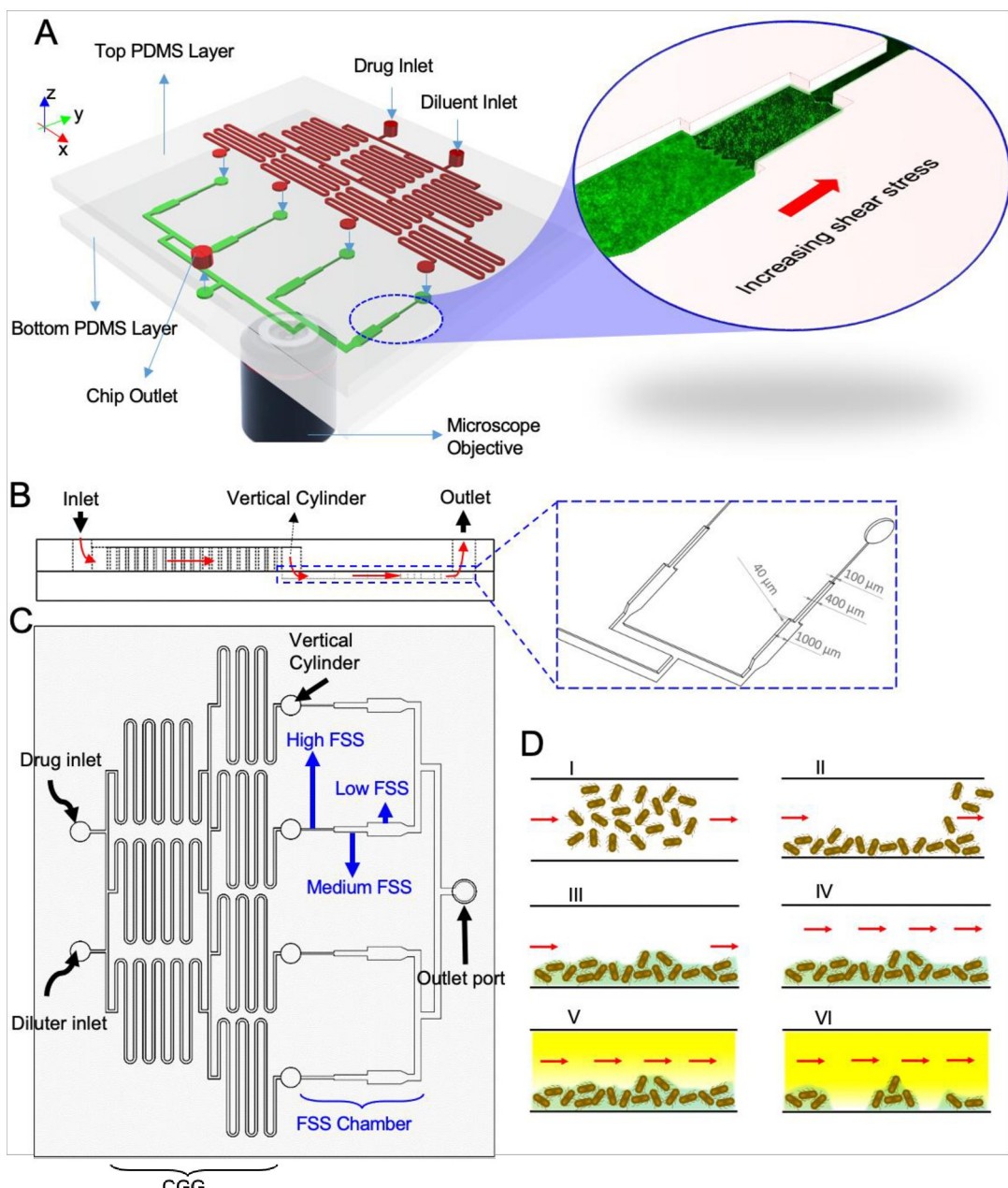

**Fig 1. Integrated microfluidic platform allows investigation of combinatory effects of chemical and fluid shear stress (FSS).** Ports and channels on the top layer are colored in red. Microchambers and collector channels on the bottom layer are colored in green. Vertical cylinders mediate the connection between the channels on the top and the bottom layer A) Schematic representation of the 2-layer physicochemical analysis biofilmchip (2PAB) B) Cross-sectional view of the platform with red arrows represent schematically the fluid flow streamline, and detailed dimensions of key features of the FSS chambers. C) Detailed schematic of the device with the concentration gradient generator (CGG) on the top layer providing stepwise antibiotic dilutions, and the FSS chambers on the bottom layer providing three zones of different shear stress levels in each chamber for testing on established biofilms. D) The workflow for biofilm formation and antibiotic/FSS treatment. Red arrows indicate the direction of fluid flow when applicable.

## Antibiotic susceptibility of planktonic bacteria

Planktonic sensitivity of *E. coli* LF82 and *P. aeruginosa* to gentamicin and streptomycin were examined in 96 well plates. Overnight cultures of bacteria were exposed at $10^5$ CFU/mL to a

range of antibiotics from 0–200 μg/mL. We visually inspected the wells after incubation to determine the planktonic minimum inhibitory concentration (MIC) for the respective bacteria/antibiotic combination. Cation adjusted Mueller Hinton broth was used as a culture medium.

## Antibiotic susceptibility of bacterial biofilm using MBEC Assay® Kit

The minimal biofilm eradication concentration (MBEC) for these bacteria/antibiotic combinations were examined with a commercially available system (MBEC Assay® Biofilm Inoculator with 96 well base, Inovotech) following manufacturer's instructions. In brief, colonies of *E. coli* LF82 or *P. aeruginosa* were picked from 24-hr grown LB agar plates and suspended in sterile PBS. The concentration of the suspended bacteria was adjusted by matching their OD600 with McFarland 0.5 standard. Then, bacteria were diluted to $10^5$ CFU/mL in 1X M9 minimal medium and used to inoculate a 96-well plate with 200 μL per well. M9 minimal medium was made of 1X M9 salts (33.7 mM $Na_2HPO_4$, 22 mM $KH_2PO_4$, 8.55 mM NaCl, and 19 mM $NH_4Cl$), plus 0.1 mM $CaCl_2$, 2 mM $MgSO_4$, 10 mM $KH_2PO_4$, 50 μM $FeSO_4$, 5 nM vitamin B1, 150 nM vitamin B12, 15 nM adenosylcobalamin, and 15 nM cobinamide dicyanide. Glucose (20 mM) was used as the carbon source. The MBEC plate and peg lid were incubated at 37 ˚C overnight (16–18 h) with shaking at 110 rpm to form biofilms, then challenged with antibiotics for 24 h. The antibiotics were prepared in a separate 96-well plate by 2-fold serial dilution with M9 minimal medium at ranges 320 μg/mL to 2.5 μg/mL for gentamicin and 800 μg/mL to 6.25 μg/mL for streptomycin, with a final volume of 150 μL per well. After the antibiotic challenge, we neutralized the antibiotics and removed the biofilms by sonicating for 30 minute following the kit's instructions. 100 μL of the sonicated biofilm suspension was added to 100 μL of cation adjusted Mueller Hinton broth, then incubated for 24 h. We checked the growth in the wells visually to determine the MBEC.

## Microfluidic device biofilm formation and evaluation

Prior to bacteria seeding, the biofilm chip was flushed with 70% ethanol solution and allowed to stand at room temperature for 5 min to sterilize. Overnight cultures of RFP-labeled *P. aeruginosa* and GFP-labeled *E. coli* LF82 were transferred to 1X M9 minimal medium for initial cell attachment and overnight growth of the biofilm on chip. M9 minimal medium was also supplemented with 50 μg/mL of kanamycin (for GFP- *E. coli*) or ampicillin (for RFP-*P. aeruginosa*) in all the subsequent experiments to maintain the fluorescent-protein plasmid. Bacterial suspension in M9 minimal culture media was introduced into the device from the outlet at 100 μL/h for 100 s. After that, all inlets and outlets were blocked off and the device was incubated for 1 h at room temperature for initial bacteria attachment. Afterward, free floating bacteria in the FSS chambers were washed out with fresh M9 minimal culture media from one of the inlets at 50 μL/h for 30 min. Then, the device was incubated at 37 ˚C overnight (16–18 h) in a water bath. The established biofilm was washed with 1X PBS for 30 min at 50 μL/h to remove any planktonic bacteria, then images at the $T_0$ time point of all the FSS zones and chambers were taken using a ZOE™ Fluorescent Cell Imager system (Bio-rad, Hercules, California) before experimental treatments. Since the length of each FSS zone is twice that of the field of view, two images were taken, using the left and right margins of the FSS zone for alignment, to account for the entire zone length. PBS and PBS with 15 μg/mL gentamicin or 200 μg/mL streptomycin (also supplemented with kanamycin or ampicillin) were added to the device from the two inlets at 300 μl/h (at each inlet port). The treatments were carried out for 24 h at room temperature at which time the final $T_{24}$ fluorescent images of all the FSS zones in all the chambers were taken.

## Image and data analysis

Fluorescent images of the biofilms before and after the 24 h treatment with antibiotics in combination with FSS were analyzed using MatLab. First, we made sure that all the images are located horizontally and that the upper and lower walls (in the x direction) of the channels were detected and surface coverage and total fluorescent intensity were calculated via the following algorithm. The images of green (GFP-*E. coli*) and red (RFP-*P. aeruginosa*) signals were read individually and converted to grayscale 8-bit images. The 2D median filter with a 5 by 5 pixel kernel was used to reduce background noise in the images. A line was used to scan the entire scope of the image horizontally and to detect the intensity change associated with the walls. Then the area outside of the channels was masked and a background intensity of 25 out of 255 was used for the minimum intensity associated with bacterial growth that also satisfies the width of the channels and is consistent among all the images of all three areas. S2 Fig in S1 File shows the intensity of gray value over a vertical line is approximately 25 at both ends of the side walls. Then the total fluorescent intensity was calculated as the sum of all the pixel intensities in the resulting image. The sum of all the pixels indicating bacteria on the surface was defined as surface coverage. Percent changes in surface area coverage and total fluorescent intensity in response to exposure to the antibiotics and FSS were calculated by dividing them against $T_0$ values within each treatment group of each experiment. Because the length of each FSS zone was twice that of the field of view of the microscope, two images were taken to encompass the entire FSS zone. The images were analyzed for the change in total fluorescent intensity or surface coverage, and the values were averaged across each of the three FSS zones.

## Statistical analysis

The experimental characterization of the 2PAB device with resazurin was conducted in triplicate and the mean (±) standard deviation was reported. The on-chip biofilm eradication experiments were conducted in three independent experiments. Significant differences in relative surface coverage and total fluorescence intensity after treatment with antibiotics and FSS were examined with two-way ANOVA with main effects of stress, concentration, and their interaction, followed by a Tukey test using JMP Pro version 16.0.0.

## Results and discussion

### Model description and working principles

The integrated biofilm chip design contains two PDMS layers that are bonded together to form the final assembled device (Fig 1A). The choice of PDMS is advantageous as the device can be used for investigating silicon coating materials for a broad range of implantation applications. PDMS is widely used as a coating material in neural and cochlear implants [40], and PDMS-based biomaterials are used for catheter fabrication with antimicrobial properties [52]. Additionally, using PDMS for this device guarantees its universality since the stiffness is mechanically tunable by several orders of magnitude. That is, to mimic the specific environment where the biofilm of interest grows, the operator can tune the stiffness of the PDMS substrate by changing the ratio of PDMS to cross-linker or the baking time and temperature. For the simplicity of this proof-of-concept study, we used a standardized PDMS mixing ratio and curing condition, but future studies can further explore the effect of surface stiffness on biofilm eradication. On the top layer, a two-step, tree-like CGG (red) serves to linearly dilute the input reagent into four concentrations (0%, 33%, 66%, 100%). The bottom layer consists of four culture chambers (green) with the narrowest chamber closest to the CGG and then getting sequentially wider (stepwise) toward the outlet port. Each of the four outputs of the CGG is

internally connected to the inlets of the FSS chambers through a vertical cylinder (represented by a blue arrow pointing down). There are 3 ports on the top layer of the system: two adjacent ports for drug and diluter injection and one outlet port through which all the flow from collector channels leaves the device. The outlet port is also utilized for the injection of the cell suspension during the bacterial seeding process. From the side view, Fluid enters from the inlets in the upper layer, through the CGG, then down to the bottom layer through the vertical cylinder, and finally leaving the device via the outlet (Fig 1B—red arrows).

Different FSSs were generated in this chip by varying the width along the culture chambers; a narrow width created a higher FSS. This stepwise expansion forms three FSS zones in each chamber. Since the flow rate is constant, with the increase in the width of the chamber, there will be a decrease in the flow velocity which leads to the decrease of FSS on the surfaces. Consequently, each FSS chamber consists of a high-FSS zone (smallest width, upstream of the flow), a medium-FSS zone (medium width, in the middle), and low-FSS zone (largest width, downstream of the flow) (Fig 1C). There are 12 distinct possible combinations between low, medium, and high FSSs with 0%, 33%, 66%, and 100% of drug concentrations.

To seed the cells, the bacterial suspension was introduced from the outlet port into the device previously treated with 70% ethanol then filled with M9 minimal culture media (Fig 1D). Both the drug and diluter inlet ports were opened to allow dislodged culture media to escape. M9 minimal culture media was selected for our study as we observed that cells attached more readily and were better at forming biofilms in the absence of nutrient rich media (S1 Fig in S1 File). After seeding, all the ports were closed, and the device was incubated at room temperature (RT) for 30 minutes to allow the cells to attach to the PDMS surface. The unattached or weakly attached cells were washed out of the device by flowing additional M9 minimal culture media, then the attached cells were incubated overnight to form biofilms in the culture chambers. The developed biofilms were gently equilibrated with PBS to further wash off any free-floating bacteria before treatment with the antibiotics. To eliminate the effects of culture media on the biofilm eradication study, such as biofilm growth [4, 38], PBS was used as the test buffer instead of the M9 minimal culture media.

The two-layer structure of the 2PAB device offers an advantage in that bacteria seeding can be done from the outlet port without the risk of seeding bacteria into the CGG's channels. It is often challenging to seed culture chambers connected to CGGs since the cell suspension can migrate into the CGG section due to the capillary effect inside the microchannels. The unique design of our chip prevents bacterial migration into the CGG because of the large radius cylindrical transition connecting the end of the CGG with the beginning of the FSS section which limits the cell seeding to the bottom layer.

The present platform is designed in a way to handle the maximum flow rate of 300 μl/h (at each inlet port) which leads to the generation of 20 dyne/cm$^2$ of FSS at maximum and ~1.5 dyne/cm$^2$ at minimum which is an acceptable range for most industrial and biomedical applications. The inputs to the microsystem are an adjusted flow rate, the initial concentration, and the antibiotic which are all set by the operator based on the desired application. All 12 states will be generated automatically by the chip based on these three input parameters.

## On-chip generation of concentration gradient and FSS

Designing and optimizing our integrated system required delicate considerations since two flow rate-sensitive interconnected systems were involved. The performance of the tree-like CGG was a function of flow rate, and CGGs lose efficiency as the flow rate increases. Consequently, flow rate was the limiting parameter in designing the CGG [44, 53]. At the same time, considering the applicability of the chip for a vast span of applications, for example a broad

range of FSS intensities from 1.5–20 dyne/cm$^2$, high flow rates will be required. As a result, the goal of the system design and optimization was to make it possible to have high FSSs while maintaining the performance of the CGG. We used COMSOL Multiphysics 5.5 to run simulations to help us choose ideal conditions. To ensure a precise and reliable design, we varied the following parameters: inlet flow rate, width and height of the CGG's channels, mixing length of CGG at each of its two stages, locations of the T-junctions between the first and the second stages of the CGG, width and height of the FSS chambers, and the architecture of the collector channels for balancing the total pressure drop. We chose to design the chip as a two-layer system because the height of the channels at each layer is independently adjustable which allows us to optimize the system. For example, because the two layers were made independently, we designed a shallower channel for the bottom layer to achieve a larger range of FSSs, whereas to maintain an acceptable performance within the CGG we made its channel height higher. By doing this, our design allows for a fixed flow rate at the inlet of the system which leads to a low-velocity flow on the top CGG layer for optimal performance of the CGG, while creating a high-velocity flow on the bottom FSS layer, which facilitates the generation of high FSS intensities.

For numerical simulation in COMSOL, the geometry of the design was meshed with triangular grids with finer resolution near the walls and surfaces for more precise solution boundary layers and calculated FSSs. Mesh studies were performed for the CGG starting from approximately 150,000 grids up to 2,529,662. The convergence criteria were chosen to be 1% of error and the final mesh skewness quality was 95%. For FSS chambers, the final mesh number was 252,480 with a skewness quality of 97% (starting from 54,040 for mesh study). Flow of the fluid is laminar in the whole system and is governed by continuity and Navier-Stokes equations. The mixing phenomenon in the CGG is also modeled with one-way coupling of laminar flow equations with the dilute species transport physics. The boundary conditions at all walls and surfaces are no-slip for flow and no-flux for mass transport. The working flow is water with the flow rate of 300 μL/h at the maximum and the diffusion coefficient of the drug was chosen to be lower than the average used for most drugs in the literature to ensure the universality of the design. All the equations and their validity for microchip design and simulation align with previously published studies on microfluidic systems [42, 44, 54, 55].

The efficiency of the CGG to generate a concentration gradient was characterized by both a computational fluid dynamics (CFD) simulation and experimental observations using the relative resazurin concentration in the culture chambers (Fig 2). The normalized concentrations of resazurin flowing into the culture chambers after exiting the CGG indicated that all mixing was done in the CGG with less than a 2 percent deviation (0%, 32.7%, 67.2%, 100%) from ideal values (0%, 33%, 66%, 100%) [44].

By increasing the width of each FSS chamber, three FSS zones with high, medium, and low shear values are formed, and the maximum FSS values in each region increase linearly with increasing flow rate (Fig 3). While the FSS is constant along each zone, at the connecting points from one zone to another a sharp drop in FSS is observed which is very small and considered negligible. However, the FSS value across the width (side to side) of each zone is not uniform. Adjacent to the walls, the biofilm experiences a lower magnitude of FSS than the average FSS experienced toward the center of the zone. Therefore, for each zone, the FSS is constant along the length of the zone, but FSS varies width-wise.

The FSS distribution on the biofilm cultured in each FSS chamber at the maximum working flow (300 μl/h from each inlet) is represented by blue dashed lines (Fig 3). Because of the large difference between the FSS range in the three zones, 0–1.5 dyne/cm$^2$ in the low FSS zone versus 0–20 dyne/cm$^2$ in the high FSS zone, we simulated the FSS profile in each zone separately to better demonstrate the non-uniformity of the FSS distribution in each zone. For the low FSS

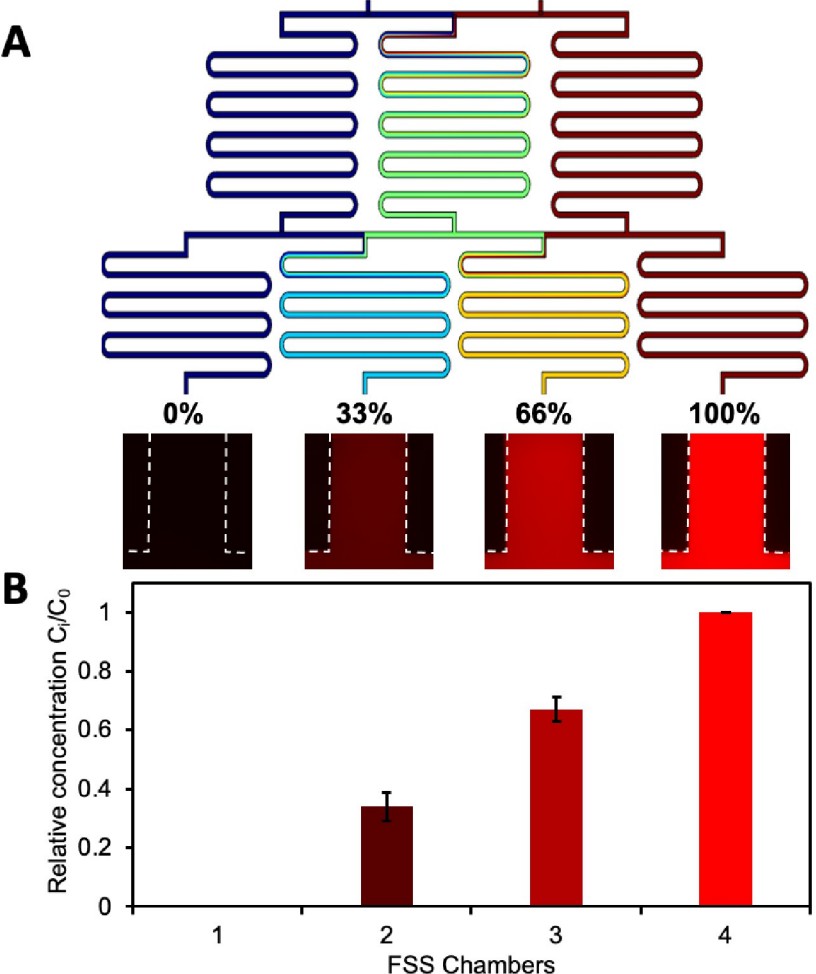

**Fig 2. Characterization of concentration gradient formation.** (A) Computational fluid dynamic (CFD) simulation of the concentration gradient generator (CGG) and (B) experimentally obtained fluorescent images of the resulting concentration gradient. Graph showing normalized concentrations of resazurin in four FSS chambers. FSS chambers 1 and 4 were set as 0 and 100 relative concentrations, and the mean relative concentration ± standard deviation of FSS chamber 2 and 3 are shown.

zone, 3% of the biofilm from each side is exposed to a FSS lower than 95% of the magnitude of the uniformly distributed FSS in the center. This 3% forms a margin where physical parameters possess a high degree of uncertainty. The rest of the biofilm (94%) experiences the same value of FSS with less than 5% deviation. For the medium FSS zone, this margin is 7% and finally for the high FSS zone it is 27%. The non-uniformity of FSS distribution in the width of all three FSS zones was calculated. We found that FSS in the medium and low FSS regimes is almost uniform while in the case of high FSS regime there is a high degree of heterogeneity (Fig 3B). The variations of FSS inside the microchannels, and the consequent margins, must be considered specifically in high-shear regime studies on cell monolayers [34] and biofilms [56] on microchips.

## On-chip effects of antibiotics on bacterial biofilms

To explore the functionality of this chip, we examined the effects of antibacterial concentration, FSS, and the combined impact of antibacterial treatment with FSS on established biofilms

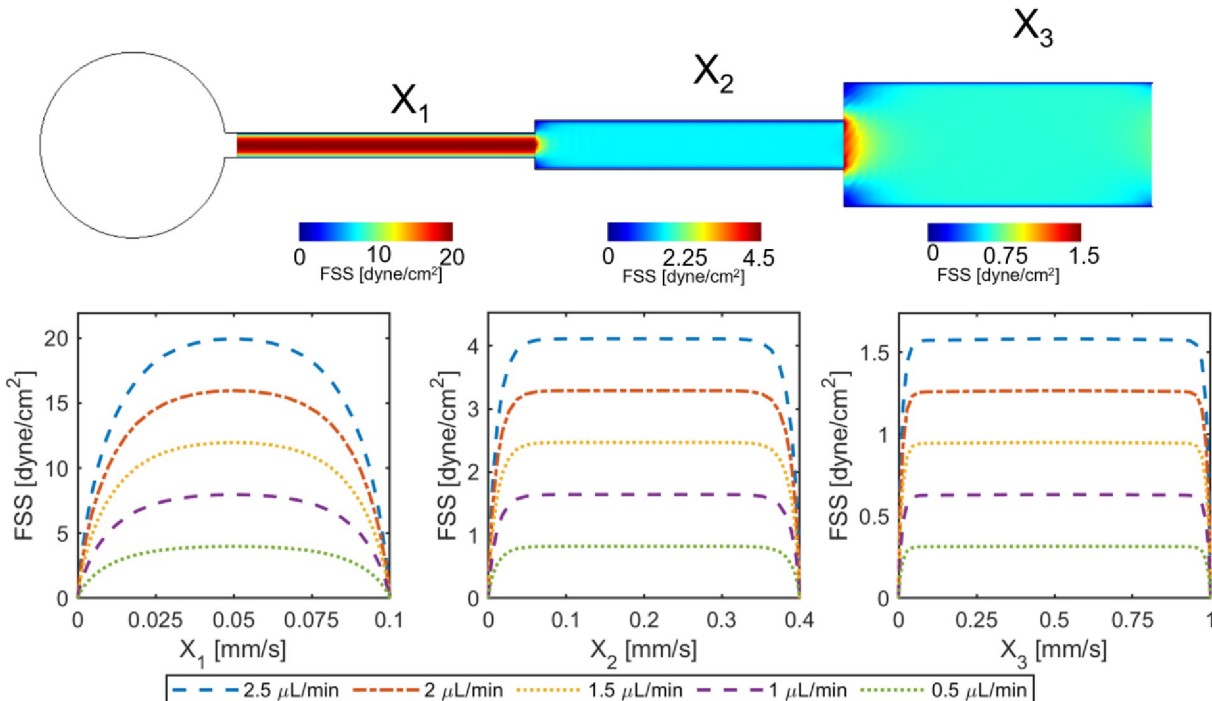

**Fig 3. Characterization of FSS distribution on cultured biofilms for each of the three zones of the expanding chamber by computational fluid dynamic simulation.** Except for a small fraction of the surface near the transitions between the zones, the FSS is uniform along the expanding chambers on the biofilm.

of *E. coli* LF82 and *P. aeruginosa* PA01. First, we experimentally determined that the planktonic MICs of *E. coli* and *P. aeruginosa* toward gentamicin (<2 μg/mL) and for streptomycin (25 μg/mL) were the same. We preliminarily screened for the minimum concentration for each antibiotic to reduce or eradicate the biofilm and found that at concentrations around 8-fold higher than the MICs, both antibiotics were able to negatively affect the *E. coli* biofilms at all FSS levels. One of the aims of this proof-of-concept study was to demonstrate the species-specificity and antibiotic-specificity that can be observed using this system. Therefore, we used the same starting concentrations of these drugs (15 μg/mL for gentamicin and 200 μg/mL for streptomycin) to test against the biofilms of both bacteria.

The fluorescent intensity from all images of the *E. coli* biofilms before and after the 24 h combined treatments with gentamicin and FSS were quantified using MATLAB, and categorized into total fluorescent intensity and surface coverage, which is the total fluorescent intensity value divided by the surface area of the analyzed region (Fig 4). The biofilms established evenly in both the high and medium FSS region, while there seems to be a slightly higher concentration of bacteria in the low FSS region (Fig 4A). We attribute this difference to the washing step which washed away more bacteria in the high and medium FSS regimes. In a follow-up study, we found that we could resolve this issue by reducing the washing flow rate. When examined closely with a microscope, we confirmed that the higher signal is due to an increase in surface coverage (increase in x and y direction) rather than a difference in thickness of the biofilm. To account for this difference in the initial biofilm, the surface coverage and total fluorescent intensity at $T_{24}$ of each treatment were compared against the values at $T_0$. Images were taken at the lower wall of the FSS region (z = 0), accounting for the majority of the signal from the biofilm there (**S3 Fig in** S1 File).

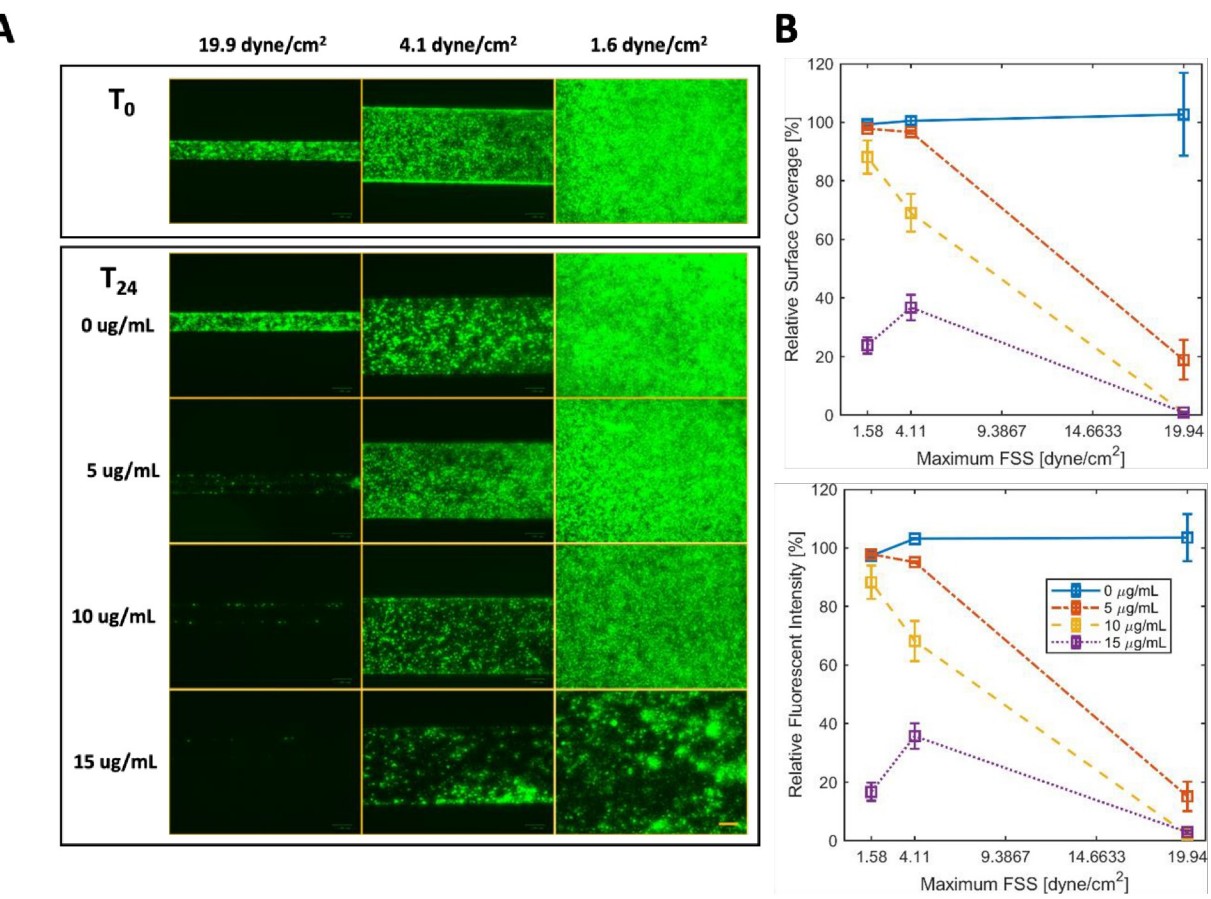

**Fig 4.** A) Fluorescent images of the on-chip E. coli LF82 biofilms and the physicochemical effects of increasing concentration of gentamicin and fluid shear stress after 24 h treatment. B) Relative changes in surface coverage and total fluorescent intensity of E. coli biofilms after 24 h treatment under different concentrations of gentamicin and levels of fluid shear stress were evaluated by MatLab analysis of fluorescent images. Data represent the mean of three replicates, reported as the percentage normalized with the initial time point ($T_0$) of each treatment. Values are expressed as means ± SEM, n = 3. Scale bar = 100μm.

After 24 h of treatment with gentamicin and FSS, we observed a decrease in the total fluorescence intensity and/or surface coverage of the *E. coli* biofilms with respect to an increase in antibiotic and FSS. The most noticeable difference can be seen in the biofilms treated with the antibiotic at the highest FSS at 19.9 dyne/cm$^2$ and the biofilms treated with the highest antibiotic concentration at 15 μg/mL. Quantitative analysis of the changes in surface coverage and total fluorescent intensity confirms this visual observation indicating that the effects of antibiotics, FSS and the combined treatment of FSS and antibiotics were statistically significant (p < 0.05) (Fig 4B). This finding is consistent with other literature reports on the effect of gentamicin on *E. coli* biofilms that were grown for 24 h [57]. This evidence supports the use of the 2PAB platform to examine the effect of antibiotics and FSS on a 24 h established biofilms.

The effect of streptomycin and FSS on the *E. coli* biofilm was studied to assess the impact of changing antibiotics. The results were similar to gentamicin, increasing antibiotic concentration and FSS alone were found to provide a statistically significant reduction of biofilm surface coverage and total fluorescent intensity (p < 0.05) (Table 1 **and S4 Fig in** S1 File). However, unlike in the gentamicin study, increasing FSS did not have a statistically significant synergistic effect on biofilm reduction.

**Table 1. Species-specific differences in the changes to surface coverage and fluorescent intensity of *E. coli* and *P. aeruginosa* biofilms after treatment with i) gentamicin and ii) streptomycin under fluid shear stresses for 24 h.**

| Gentamicin concentration/ FSS | | *E. coli* | | | *P. aeruginosa* | | |
|---|---|---|---|---|---|---|---|
| | | 19.9 dyne/cm$^2$ | 4.1 dyne/cm$^2$ | 1.6 dyne/cm$^2$ | 19.9 dyne/cm$^2$ | 4.1 dyne/cm$^2$ | 1.6 dyne/cm$^2$ |
| Relative total surface coverage (%) | 0 µg/mL | 102.72 ± 28.31[a] | 105.05 ± 3.24[a] | 100.15 ± 0.54[a] | 104.31 ± 4.91[a] | 99.20 ± 1.21[a] | 93.56 ± 3.08[ab] |
| | 5 µg/mL | 18.89 ± 13.67[cd] | 96.62 ± 1.02[a] | 97.91 ± 0.80[a] | 90.01 ± 5.16[ab] | 77.30 ± 13.84[ab] | 97.77 ± 3.77[ab] |
| | 10 µg/mL | 0.89 ± 0.30[d] | 69.07 ± 12.83[abc] | 88.08 ± 11.36[ab] | 74.56 ± 12.04[ab] | 57.96 ± 17.51[ab] | 72.50 ± 3.32[ab] |
| | 15 µg/mL | 0.93 ± 0.35[d] | 36.80 ± 8.70[bcd] | 23.78 ± 5.62[cd] | 44.58 ± 12.78[b] | 50.82 ± 18.88[ab] | 62.12 ± 9.86[ab] |
| Relative total fluorescent intensity (%) | 0 µg/mL | 127.97 ± 39.09[a] | 110.04 ± 8.98[a] | 96.37 ± 0.18[ab] | 94.83 ± 4.78[a] | 126.62 ± 13.83[a] | 103.78 ± 4.60[a] |
| | 5 µg/mL | 15.11 ± 9.97[c] | 95.14 ± 0.70[ab] | 97.86 ± 0.81[ab] | 125.41 ± 26.55[a] | 131.60 ± 33.38[a] | 96.97 ± 8.49[a] |
| | 10 µg/mL | 6.22 ± 0.63[c] | 59.94 ± 13.73[ab] | 77.92 ± 11.23[ab] | 74.93 ± 14.08[a] | 52.64 ± 18.23[a] | 87.50 ± 21.82[a] |
| | 15 µg/mL | 0.93 ± 1.20[c] | 36.80 ± 8.72[bc] | 23.78 ± 6.17[c] | 67.18 ± 2.82[a] | 50.55 ± 18.07[a] | 65.59 ± 12.64[a] |
| Streptomycin concentration/ FSS | | *E. coli* | | | *P. aeruginosa* | | |
| | | 19.9 dyne/cm$^2$ | 4.1 dyne/cm$^2$ | 1.6 dyne/cm$^2$ | 19.9 dyne/cm$^2$ | 4.1 dyne/cm$^2$ | 1.6 dyne/cm$^2$ |
| Relative total surface coverage (%) | 0 µg/mL | 90.70 ± 20.78[a] | 88.55 ± 5.72[a] | 98.87 ± 0.64[a] | 107.97 ± 3.38[a] | 98.22 ± 0.87[a] | 96.99 ± 3.08[a] |
| | 66 µg/mL | 32.30 ± 16.15[abc] | 82.26 ± 16.83[ab] | 94.72 ± 3.34[a] | 96.57 ± 5.26[a] | 96.57 ± 1.46[a] | 99.96 ± 0.07[a] |
| | 133 µg/mL | 5.26 ± 4.31[c] | 31.28 ± 4.14[abc] | 61.65 ± 23.87[abc] | 79.05 ± 19.79[a] | 107.41 ± 4.01[a] | 99.59 ± 0.09[a] |
| | 200 µg/mL | 13.56 ± 2.45[bc] | 15.99 ± 9.05[bc] | 49.69 ± 24.41[abc] | 103.56 ± 13.96[a] | 100.38 ± 2.36[a] | 96.85 ± 2.97[a] |
| Relative total fluorescent intensity (%) | 0 µg/mL | 74.42 ± 36.13[abc] | 88.71 ± 7.16[ab] | 99.57 ± 1.25[a] | 92.21 ± 5.27[a] | 127.42 ± 13.08[a] | 103.33 ± 4.71[a] |
| | 66 µg/mL | 28.11 ± 14.01[abc] | 77.19 ± 13.03[abc] | 94.67 ± 3.35[a] | 99.12 ± 11.04[a] | 92.10 ± 30.60[a] | 90.59 ± 6.51[a] |
| | 133 µg/mL | 7.68 ± 3.87[c] | 26.54 ± 4.05[abc] | 57.59 ± 23.89[abc] | 63.27 ± 5.86[a] | 81.44 ± 15.87[a] | 98.41 ± 20.40[a] |
| | 200 µg/mL | 11.35 ± 2.66[bc] | 12.49 ± 6.48[bc] | 56.54 ± 26.03[abc] | 93.73 ± 12.32[a] | 106.14 ± 7.58[a] | 105.98 ± 19.46[a] |

Data represent the mean of three replicates, reported as the percentage normalized with the initial time point (T$_0$) of each treatment. Values are expressed as means ± SEM, $n = 3$.

[a, b, c, d] indicate significant differences within the relative surface coverage (%) or total fluorescence intensity (%) each bacteria/antibiotic treatment group with all the FSS levels $p < 0.05$.

In contrast to the results, we found for our studies with *E. coli* biofilms, we did not observe any statistically significant effects of either antibiotic with FSS on *P. aeruginosa* biofilms, with one notable exception where increasing the concentration of gentamicin reduced the surface coverage and fluorescent intensity ($p < 0.05$). This observation suggests that the susceptibility of biofilms to antibiotics is species-specific and drug-specific [46, 58]. One of the main factors contributing to this phenomenon is the difference in the EPS composition and physicochemical characteristics between different bacterial species and even different strains of the same species, leading to differences in drug penetration rate and drug interactions [46, 47]. Another factor could be the difference in response of specific bacteria to shear stress, as *P. aeruginosa* has been shown to exhibit higher adhesion under shear stress [59].

Overall, these findings reiterate the need to experimentally determine the susceptibility of biofilms to chemical and physical treatments since the planktonic MICs is not a reliable indicator for efficacy with biofilms. Furthermore, we compared the results obtained using our platform and the MBEC results obtained using the commercially available gold standard method (MBEC Assay® Kit) and found that the MBEC values by the conventional method reflect the findings in treatments at the lowest fluid shear stress (1.6 dyne/cm$^2$) of our device. The MBEC values of *E. coli* treated with gentamicin and streptomycin were 10 and 200 µg/mL, and the values of *P. aeruginosa* treated with these antibiotics were 20 and 400 µg/mL, respectively. While a useful comparison, these values, however, reflect different phenomena, specifically where the commercial assay calls for sonication to disrupt the remaining biofilm after an antibiotic

challenge, whereas our system quantifies based on the remaining intact biofilm on the surface. Additionally, while the conventional method focuses more on antibiotic penetration and killing of the bacteria within the biofilm, our method allows further examination of other properties of the biofilm such as ability to resist shear stress or adhesion/detachment from the surface. Additional future studies will provide increased insight into the differences between these methods and observations, and how they might apply to real-world applications of eradicating biofilms.

## Conclusion

We have designed a 2-layer physicochemical analysis biofilmchip (2PAB) platform for high-throughput examination of physicochemical effects on 24 h-established biofilms. The double-layer design of our device allowed for a universal CGG and three FSS regimes to be integrated into the same chip. This produced a chip that has a broad range of applications, which can test 12 distinct combinations of chemical concentrations and FSS simultaneously. We demonstrated experimentally, and using computer simulation, the efficiency and reliability of the CGG to dilute antibiotics into four linear concentrations and that the FSS chambers generated three FSS levels by stepwise expansion of the channel width. Finally, we showed proof-of-concept of the 2PAB device by examining the effects of gentamicin and streptomycin on the on-chip established biofilms of *E. coli* LF82 and *P. aeruginosa*. We observed concentration and FSS dependent reductions in the surface coverage and total fluorescence intensity of *E. coli* biofilms after 24 h of treatment. In addition, notable species-specific differences in efficacy of the tested antibiotics when compared to treatment of *P. aeruginosa* biofilms were also observed in our study. These results suggest that the 2PAB device is a useful platform in studying bacterial biofilm responses to physicochemical factors and determining the parameters for an effective biofilm eradication strategy. Furthermore, we believe that the platform has potential as a high-throughput screening tool with versatile applications. The high-throughput screening ability of the platform can be improved by adding more FSS zones and/or having larger CGG to achieve a broader range of FSS and more increments of both FSS levels and antibiotic concentrations. Here, we demonstrated only 12 combinations (4 concentrations vs. 3 shear stress levels), but the platform can be designed to include a higher number of factors (for example, 12 concentrations vs. 12 shear stress levels) using the same design and optimization principles. The platform can offer testing at as many concentration levels as a commercially available system based on 96-well plate, with the addition of fluid shear stress factor.

## Supporting information

**S1 File.**
(DOCX)

## Acknowledgments

The authors thank Dr. Kelley Donaghy for scientific writing and editing support.

## Author Contributions

**Conceptualization:** Ann V. Nguyen, Arash Yahyazadeh Shourabi, Shiying Zhang, Kenneth W. Simpson, Alireza Abbaspourrad.

**Data curation:** Ann V. Nguyen, Arash Yahyazadeh Shourabi, Mohammad Yaghoobi.

**Formal analysis:** Ann V. Nguyen, Arash Yahyazadeh Shourabi, Mohammad Yaghoobi.

**Funding acquisition:** Alireza Abbaspourrad.

**Investigation:** Ann V. Nguyen, Arash Yahyazadeh Shourabi, Mohammad Yaghoobi.

**Methodology:** Ann V. Nguyen, Arash Yahyazadeh Shourabi, Mohammad Yaghoobi, Shiying Zhang.

**Project administration:** Alireza Abbaspourrad.

**Resources:** Kenneth W. Simpson, Alireza Abbaspourrad.

**Supervision:** Alireza Abbaspourrad.

**Writing – original draft:** Ann V. Nguyen, Arash Yahyazadeh Shourabi, Mohammad Yaghoobi, Shiying Zhang.

**Writing – review & editing:** Ann V. Nguyen, Arash Yahyazadeh Shourabi, Mohammad Yaghoobi, Shiying Zhang, Kenneth W. Simpson, Alireza Abbaspourrad.

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
