## [Decision Letter · Decision Letter 0]

7 Jun 2022

PONE-D-22-12144A high-throughput integrated biofilm-on-a-chip platform for the investigation of combinatory physicochemical responses to chemical and fluid shear stressPLOS ONE

Dear Dr. Alireza Abbaspourrad,

Thank you for submitting your manuscript to PLOS ONE. After careful consideration, we feel that it has merit but does not fully meet PLOS ONE’s publication criteria as it currently stands. Therefore, we invite you to submit a revised version of the manuscript that addresses the points raised during the review process.

We look forward to receiving your revised manuscript.

Kind regards,

Abdelwahab Omri, Pharm B, Ph.D, Laurentian University

Academic Editor

PLOS ONE

Journal Requirements:

2. Please ensure you have stated in the Methods section of your manuscript text the origin (supplier or manufacturer) of the materials and reagents used in this study.

Additionally, please note that PLOS ONE has specific guidelines on code sharing for submissions in which author-generated code underpins the findings in the manuscript. In these cases, all author-generated code must be made available without restrictions upon publication of the work. Please review our guidelines at https://journals.plos.org/plosone/s/materials-and-software-sharing#loc-sharing-code and ensure that your code is shared in a way that follows best practice and facilitates reproducibility and reuse.

Reviewers' comments:

Reviewer's Responses to Questions

**Comments to the Author**

1. Is the manuscript technically sound, and do the data support the conclusions?

Reviewer #1: Partly

Reviewer #2: Partly

2. Has the statistical analysis been performed appropriately and rigorously? 

Reviewer #1: I Don't Know

Reviewer #2: Yes

3. Have the authors made all data underlying the findings in their manuscript fully available?

Reviewer #1: No

Reviewer #2: No

4. Is the manuscript presented in an intelligible fashion and written in standard English?

Reviewer #1: Yes

Reviewer #2: Yes

5. Review Comments to the Author

Reviewer #1: This is a very interesting piece of work that describes a very promising technique for the study of biofilm eradication. I am not qualified to make a judgement on the fluid dynamics of the device. However, I do have a issues with the methodology. I think that if they are addressed, then the to manuscript will be greatly improved:

1) I cannot see a description of the strain of P. aeruginosa. Why was the particular strain chosen? This must be included in the Experimental section.

2) The choice of PDMS for the device is very interesting. The authors should describe the choice of PDMS in more detail, highlighting its use in current medical devices, e.g. catheters. What are the properties of this material with regard to bacterial adhesion and growth? Do the bacteria adhere mainly to the glass bottom layer or the PDMS? If to the glass, what is the relevance with regard to the clinical situation that the authors appear to be addressing in this application of the device?

3) A MIC is performed but an MBEC (i.e. biofilm eradication) under standard conditions would have been very useful, to compare the type of data most often obtained with the data obtained using the authors' device.

4) It is unclear from the Experimental as to exactly how images were obtained. What type of micrscope was employed? Was a z-stack made after locating the upper and lower walls of the channel? If not, how do the authors' guarantee that the fluorescence truly represents the three-dimensional array of bacteria? Was the depth of field somehow large enough to have the whole of the biofilm in focus? How does out-of-focus fluorescence register compared to in-focus? Is there a bias when measuring a thicker biofilm compared to a thinner one?

5) How was the background fluorescence of 25/255 calculated? Did background not vary at all in different channels?

6) It is unclear in Figure 1 as to where the microscope is positioned - can this be clarified? In addition, what magnification was used? Was the whole 'zone' of each channel imaged just once at each time point?

7) It would be very helpful to include an indication of the statistical significances observed in Table 1 (or as a separate Table, if necessary), rather than merely mentioning some examples in the text.

8) The authors conclude that the different susceptibilities of biofilms to antibiotics is a result mainly of drug penetration etc due to 'the difference in EPS composition and physicochemical characteristics'. An alternative explanation might be the response of P. aeruginosa to shear stress ( doi: 10.1016/j.bpj.2010.11.078). Increasing adhesion of the biofilm to the substrate might counteract the effects of the antibiotics.

Reviewer #2: Reviewer comments

Article: A high-throughput integrated biofilm-on-a-chip platform for the investigation of combinatory physicochemical responses to chemical and fluid shear stress

Manuscript ID: PONE-D-22-12144.

The manuscript describes the use of integrated double-layered microfluidic chip for assessing the physiochemical responses of bacterial biofilms or to assess the effectiveness of biofilm removal methods. Yet, it requires strenuous experimental validation to prove the same.

Major comments

1. The proof-of-concept study revealed the potential of combinatorial effect of antibiotics and fluid shear stress. Yet, the study is limited to specific bacterial pathogens and antibiotics. As a proof-of-concept study, studying its effect on different pathogens and with last resort antibiotics is much needed. More, the feasibility of this chip-based high-throughput screening method should be vigourously discussed based on the results obtained from the results.

2. Showing the biofilm forming ability of the test pathogen used in this study as supplement would able the readers to interpret the data.

3. Like figure 4 and supplemental file, showing the data for biofilm eradication of Pseudomonas aeruginosa (fluorescent images) in the presence of FSS and antibiotics would be better as supplement.

4. Page 16, the MIC of gentamycin (2 ug/ml) and streptomycin (25 ug/ml) was same for both the pathogen. For the study using fluidic chip, the concentration used were 15 and 200 ug/ml, which is more than 10-fold higher than the MIC. Justify?

6. PLOS authors have the option to publish the peer review history of their article (what does this mean?). If published, this will include your full peer review and any attached files.

Reviewer #1: **Yes: **Peter Monk

Reviewer #2: No

---

## [Author Response · Author response to Decision Letter 0]

14 Jul 2022

We have attached a response document in the files section that addresses all of the reviewer comments. We are very thankful for their careful attention, their suggestions and comments strengthened our manuscript.

---

## [Editor Report · Decision Letter 1]

18 Jul 2022

A high-throughput integrated biofilm-on-a-chip platform for the investigation of combinatory physicochemical responses to chemical and fluid shear stress

PONE-D-22-12144R1

Dear Dr. Alireza Abbaspourrad,

We’re pleased to inform you that your manuscript has been judged scientifically suitable for publication and will be formally accepted for publication once it meets all outstanding technical requirements.

Kind regards,

Abdelwahab Omri, Pharm B, Ph.D, Laurentian University, Canada

Academic Editor

PLOS ONE

---

## [Editor Report · Acceptance letter]

4 Aug 2022

PONE-D-22-12144R1 

A high-throughput integrated biofilm-on-a-chip platform for the investigation of combinatory physicochemical responses to chemical and fluid shear stress 

Dear Dr. Abbaspourrad:

I'm pleased to inform you that your manuscript has been deemed suitable for publication in PLOS ONE. Congratulations! Your manuscript is now with our production department. 

Kind regards, 

on behalf of

Dr. Abdelwahab Omri 

Academic Editor

PLOS ONE